# Rapid Response of Refractory Systemic Lupus Erythematosus Skin Manifestations to Anifrolumab—A Case-Based Review of Clinical Trial Data Suggesting a Domain-Based Therapeutic Approach

**DOI:** 10.3390/jcm11123449

**Published:** 2022-06-15

**Authors:** Marlene Plüß, Silvia Piantoni, Chris Wincup, Peter Korsten

**Affiliations:** 1Department of Nephrology and Rheumatology, University Medical Center Göttingen, 37075 Göttingen, Germany; marlene.pluess@med.uni-goettingen.de; 2Rheumatology and Clinical Immunology Unit, Department of Clinical and Experimental Sciences, ASST Spedali Civili and University of Brescia, 25121 Brescia, Italy; slv.piantoni@gmail.com; 3Department of Rheumatology, King’s College Hospital, London SE5 9RS, UK; c.wincup@ucl.ac.uk

**Keywords:** systemic lupus erythematosus, anifrolumab, interferon, cutaneous lupus erythematosus

## Abstract

Systemic lupus erythematosus (SLE) is a clinically heterogeneous autoimmune disease, and organ manifestations, such as lupus nephritis (LN) or skin disease, may be refractory to standard treatment. Therefore, new agents are required to allow for a more personalized therapeutic approach. Recently, several new therapies have been approved internationally, including voclosporine for LN and anifrolumab for moderately to severely active SLE. Here, we report a case of SLE with a predominant and refractory cutaneous manifestation despite combination treatment with glucocorticoids, hydroxychloroquine, mycophenolate mofetil, and belimumab, which had been present for more than 12 months. Belimumab was switched to anifrolumab, and the patient responded quickly after two infusions (eight weeks) with a reduction in the Cutaneous Lupus Assessment and Severity Index (CLASI) from 17 to 7. In addition, we review the available clinical trial data for anifrolumab with a focus on cutaneous outcomes. Based on phase II and III clinical trials investigating the intravenous administration, a consistent CLASI improvement was observed at 12 weeks. Interestingly, in a phase II trial of subcutaneous anifrolumab application, CLASI response was not different from placebo at 12 weeks but numerically different at 24 and 52 weeks, respectively. Thus, anifrolumab emerges as an attractive new therapeutic option suggesting a possible domain-based approach.

## 1. Introduction

Anifrolumab (ANI), a fully human monoclonal antibody directed against the type I interferon (IFN) receptor subunit 1, has recently been approved as add-on therapy for moderately to severely active systemic lupus erythematosus (SLE) based on the results of two phase III trials (TULIP-1 and TULIP-2) [1,2]. There is limited experience in clinical practice outside of a trial setting, but an early access program was available in Germany until March 2022. Here, we report the first use of ANI in an SLE patient with refractory cutaneous manifestations outside a clinical trial setting and review the data of ANI clinical trials focusing on the cutaneous domain.

## 2. Case Description

The patient is a 30-year-old female with a 13-year history of SLE based on acute cutaneous lupus, polyarthritis, positivity for antinuclear antibodies (abs), anti-double-stranded DNA, anti-Ro/SSA, anti-U1-snRNP abs, and complement consumption. Over the years, additional findings included photosensitivity, class II lupus nephritis (LN), and positive anti-phospholipid abs.

She presented in October 2021 to her routine visit and complained of worsening skin lesions and joint pain over the preceding three months (Figure 1A–C).

Her skin lesions consisted of non-pruritic erythematous lesions distributed symmetrically over the upper trunk and back, as well as the face, arms, and hands. There were no areas of scarring. Her skin rash was most consistent with subacute cutaneous lupus erythematosus (SCLE). The Cutaneous Lupus Erythematosus Disease Area and Severity Index (CLASI) was determined with a score of 17.

The metacarpophalangeal II to V and proximal interphalangeal joints II to V were slightly swollen and tender to palpation. Anti-dsDNA abs were elevated at 114 IU/mL (normal range [NR] <15 IU/mL), C3 was 0.71 g/L (NR 0.82–1.93 g/L), and C4 was 0.08 g/L (NR 0.15–0.57 g/L). The Systemic Lupus Erythematosus Disease Activity Index 2000 (SLEDAI-2K) was calculated as 10.

Previous therapy included hydroxychloroquine (200 mg/day), azathioprine, and varying doses of prednisolone. Three courses of medium to high oral prednisolone doses (0.5–1 mg/kg of body weight) to control her skin disease and flares of polyarthritis were given over the previous six months.

Her current SLE treatment included hydroxychloroquine (200 mg/day), low-dose prednisolone (5 mg/day), mycophenolate mofetil (500 mg twice daily, higher doses not tolerated), and subcutaneous belimumab (BEL) (200 mg/week), which had been started more than 12 months before. In this refractory patient with active cutaneous and joint disease, BEL was switched to ANI.

Anifrolumab treatment was initiated in January 2022. Eight weeks later, after receiving two intravenous infusions of 300 mg four weeks apart, the skin lesions had improved significantly (Figure 1D–F).

In addition, her complement levels and anti-dsDNA antibody titer improved moderately (Figure 2A,B). The CLASI improved from 17 to 7 (Figure 2C). Joint pain and swelling also improved with treatment. No side effects or infusion reactions occurred, and the patient has lowered her prednisolone dose to 2 mg/day. After four infusions, the patient reduced her dose of mycophenolate mofetil to 500 mg/day. Follow-up is ongoing.

## 3. Review of Anifrolumab Mechanism of Action and Clinical Trials

### 3.1. Development and Mechanism of Action of Anifrolumab

The pathogenesis of SLE, which is considered a prototypic autoimmune disease, is complex and involves a myriad of immune mechanisms and various cell types [3] (Figure 3). In brief, environmental (e.g., ultraviolet radiation), viral (e.g., Ebstein–Barr virus), and hormonal triggers lead to an increased rate of apoptosis in an (epi)genetically susceptible individual [3,4]. Autoreactive B and T cells process this increased number of antigens, leading to autoantibody and immune complex formation [4]. As a result, there is an increased production of type 1 interferons (IFNs) by plasmacytoid dendritic cells (pDCs), which is a central pathogenic process [5,6]. Type 1 interferons maintain an increased autoantibody production through an autocrine loop, further activating B cells, which undergo class switching [4].

Recently, an increasing number of clinical trials, including the TULIP trials, stratified patients according to their IFN gene expression status (high vs. low) [7]. However, this has not been adopted for routine clinical practice. In view of IFNs as a key mediator in SLE pathogenesis, targeting IFNs by monoclonal antibodies (mAbs) is an appealing approach. Sifalimumab and rontalizumab, two other mAbs targeting IFN alpha, have yielded mixed results in phase II trials [8,9], and have not been developed and tested in phase III trials.

**Figure 3 jcm-11-03449-f003:**
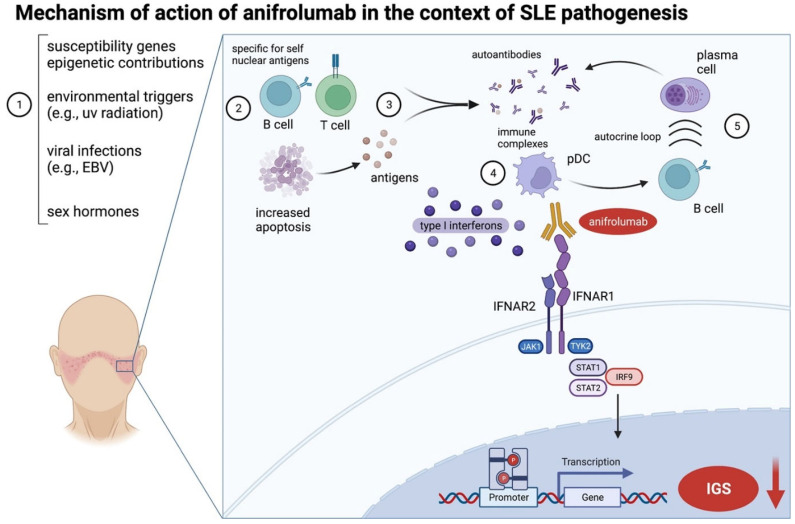
Mechanism of action of anifrolumab in the context of the hypothesized systemic lupus erythematosus (SLE) pathogenesis [3,4,6]. Genetic, epigenetic, environmental, and hormonal factors (**1**) lead to an increased rate of apoptosis. Autoreactive B and T cells specific for self-nuclear antigens recognize and process these antigens (**2**), which, in turn, leads to autoantibody and immune complex generation (**3**). Toll-like receptor signaling in B cells and pDCs (not shown) results in increased levels of type 1 interferons, mainly produced by pDCs (**4**). Type 1 interferons further stimulate B cells in an autocrine loop, and B cells exhibit class switching, which leads to a persistent production of autoantibodies (**5**). Anifrolumab binds to IFNAR1, thus inhibiting dimerization and subsequent intracellular signaling mechanisms mediated by STAT1/2 and IRF9. The net result is a decreased transcription of proinflammatory genes (the so-called interferon-gene signature, IGS) in cells of both the innate and adaptive immune systems [10]. Created with biorender.com. EBV, Ebstein–Barr virus; IFNAR, interferon receptor subunit; IGS, interferon gene signature; IRF9, interferon regulatory factor 9; JAK, Janus kinase; pDC, plasmacytoid dendritic cells; STAT, signal transducers and activators of transcription; TYK, tyrosine kinase; UV, ultraviolet.

Anifrolumab is a fully human, effector-null monoclonal antibody directed against the type I interferon (IFN) receptor subunit 1 (IFNAR1) [10]. It was engineered with mutations inserted in the heavy chain with the aim of reducing Fcγ receptor (FcγR)-mediated effector functions, such as antibody-dependent cell-mediated cytotoxicity and complement-dependent cytotoxicity [11], ultimately improving efficacy and reducing resistance through internalization by FcR [12].

Further, it has been shown that ANI promotes IFNAR1 internalization, thus blocking downstream signaling, such as signal transducers and activators of transcription 1 (STAT1) phosphorylation [10]. Finally, ANI reduces the type I IFN autoamplificaton loop sustained by pDCs [10,13].

The mechanism of action of ANI in the context of a proposed model of SLE pathogenesis is shown in Figure 3.

### 3.2. Clinical Trial Data

In this section, we will review the available clinical trial data from phase I–III clinical trials of ANI, which resulted in the approval for the treatment of moderately-to-severely active SLE in addition to standard therapy. Figure 4 shows a timeline of major clinical trials and approval dates of ANI in non-renal SLE. Of note, a phase II clinical trial in LN has been published [14]. However, we will not further analyze this trial since the focus of this review is the cutaneous domain.

#### 3.2.1. Early Phase I and Phase II Trials

Interestingly, ANI, then termed MEDI-546, was first tested in a phase I clinical trial in Systemic Sclerosis (SSc) patients [15]. In this phase I trial, 34 subjects received MEDI-546 in a dose-escalation fashion for 12 weeks. A total of 68.9% of subjects experienced mild adverse events (AEs), and 27.7% experienced moderate AEs. In addition, there were four serious AEs (skin ulcer, osteomyelitis, vertigo, and chronic myelogenous leukemia). Only the latter was judged as possibly treatment-related [15].

Since interferon signaling pathways involved in the pathogenesis of SSc and SLE share similarities [16], MEDI-546, later renamed ANI, was further investigated in a phase IIb trial in non-renal SLE [17]. In this trial, 305 participants with moderate-to-severely active SLE were randomized to receive one of two doses of ANI (300 vs. 1000 mg every four weeks for 48 weeks) or a placebo (PBO). Patients were randomized based on disease activity (SLE disease activity index-2000 [SLEDAI-2K] >10 vs. <10), glucocorticoid (GC) dose (>10 vs. ≤10 mg/day), and type I interferon gene expression (high vs. low). The SRI4 endpoint was met by more patients treated with ANI (34.3% of 99 patients for 300 mg and 28.8% of 104 for 1000 mg) compared to PBO (17.6% of 102 patients) (Table 1). With these encouraging results, two phase III trials were performed subsequently.

#### 3.2.2. Phase III Trials—TULIP-1 and TULIP-2

In the phase III trial TULIP-1, ANI 150 mg or 300 mg were compared to PBO. TULIP-1 randomized 457 patients; the primary endpoint systemic lupus erythematosus responder index-4 (SRI-4) was assessed at 52 weeks. There were no statistically significant differences in patients receiving 300 mg of ANI compared to PBO regarding this outcome measure (36% vs. 40%, respectively). Since the primary endpoint was not met, no statistical testing was performed as per the prespecified study analysis plan. However, the British Isles Lupus Assessment Group-based composite lupus assessment (BICLA), another robust outcome measure used for SLE, was numerically different (37% vs. 27% responders for ANI 300 mg vs. PBO) [2].

Therefore, the TULIP-2 clinical trial used the BICLA as the primary outcome measure [1]. TULIP-2 randomized 365 patients to ANI 300 mg or PBO. At 52 weeks, there was a statistically significant difference in the BICLA response in favor of ANI 300 mg (47.8% vs. 31.5%). These results finally led to ANI’s approval by the Federal Drug Administration (FDA) in 2021 and the European Medicines Agency (EMA) in early 2022. The different results of TULIP-1 and TULIP-2 regarding their primary efficacy measures have been discussed widely [18,19,20]. Table 1 gives an overview of the main published clinical trials and the primary outcomes.

**Table 1 jcm-11-03449-t001:** Overview of clinical trials of anifrolumab in non-renal Systemic Lupus erythematosus.

First Author, Year	Trial Acronym	Phase	N of Participants	Primary Endpoint Assessment	Outcome Measures (% Responders) ^§^ (Anifrolumab 300 mg/Placebo)		
**Intravenous administration**					**SRI-4**	**BICLA**	**CLASI**
Furie, 2017 [17]	MUSE	IIb	305	24 weeks	**34.3/17.6**	53.5/25.7	63/30.8
Furie, 2019 [2]	TULIP-1	III	457	52 weeks	**36/40**	37/27	42/25 *
Morand, 2020 [1]	TULIP-2	III	365	52 weeks	55.5/37.3	**47.8/31.5**	49/25 *
**Subcutaneous administration**							
Bruce, 2021 [21]	-	II	36	12 weeks	-/-	-/-	45/44 *

^§^ Bold indicates the primary outcome measure. Outcome measures: BICLA, British Isles Lupus Assessment Group-based composite lupus assessment; CLASI, cutaneous lupus erythematosus disease area and severity index; SRI-4, systemic lupus erythematosus responder index-4. * at 12 weeks.

## 4. Discussion

To our knowledge, this is the first description of therapy with ANI for refractory cutaneous manifestations outside a clinical trial, demonstrating very quick clinical efficacy. It is unclear which primary outcome measure is best for trials in SLE [19]. Both SRI, used in TULIP-1 [2], and the BICLA, used in TULIP-2 [1], are robust measures of treatment response. They consist of different domains and give more weight to the SLEDAI response (SRI), or BILAG domains (BICLA), respectively. Both measures do not allow for any domain to worsen. In TULIP-1, there was no difference in the CLASI between ANI and PBO; in TULIP-2, a statistically significant difference was found at week 12 [1]. A post-hoc analysis of pooled data in patients with CLASI ≥10 at baseline confirmed these results [22].

In all published phase II and III clinical trials, the Cutaneous Lupus erythematosus disease area and severity index (CLASI) was used to assess changes in skin manifestations. The CLASI aims to distinguish between activity and damage [23]. In the activity domain, erythema is graded from 0 (absent) to 3 (dark red; purple/violaceous/crusted/hemorrhagic) in different body areas. Likewise, scales/hypertrophy are judged from 0 (absent) to 2 (verrucous/hypertrophic). Further, lesions of the mucous membranes are searched for. Lastly, alopecia is assessed as present or absent. If present, the scalp is divided into four quadrants and scored, ranging from 0 (absent) to 3 (focal or patchy in more than one quadrant). To analyze the damage, various lesions are scored: First, dyspigmentation is documented as 0 (absent) or 1 (present). Next, scarring/atrophy/panniculitis is scored, ranging from 0 (absent) to 2 (severely atrophic scarring or panniculitis). Then, the duration of dyspigmentation is considered (more or less than 12 months). Finally, scarring of the scalp is scored as 0 (absent), 3 (present in one quadrant), 4 (present in two quadrants), 5 (three quadrants), or 6 (affects the whole skull). The overall score ranges from 0 to 70, and higher scores indicate more severe skin disease.

It must be noted that the CLASI response was defined as an improvement of at least 50% in participants with a minimum score ≥10. In the MUSE phase IIb trial, 77 (25%) of patients fulfilled this definition [17]. The percentage of CLASI responders at 24 weeks was 63% for 300 mg of ANI vs. PBO (30.8%), and responses were seen early on (around 50% at eight weeks) with a plateau of 60–65% response rates around week 20. In TULIP-1, the CLASI response followed the same definition, and there were 42% vs. 25% of responders favoring ANI 300 mg at 12 weeks [2]. However, the difference between ANI and PBO evened out at the end of the trial. Finally, TULIP-2 reported a CLASI responder rate of 49% vs. 25% at 12 weeks, which was maintained through 52 weeks [1].

Lastly, a phase II of subcutaneous administration of ANI in 36 patients showed no numerical differences in the CLASI response at 12 weeks (45% vs. 44%) [21]. Nevertheless, unlike the TULIP trials, the response rates steadily increased from 82% vs. 50% at 24 weeks to 91% vs. 44% at 52 weeks. The number of subjects was small, and this phase II trial was also not designed to assess any differences in the CLASI response. One possible explanation for the steadier increase compared to the rapid rise in response rates with the intravenous administration may be the slower absorption and biological efficacy following a subcutaneous application.

Furthermore, it has been shown that IFN signaling has a central role in SLE skin pathology as the IFN signature correlates with cutaneous disease activity in SLE [24], and IFN pathways contribute to enhancing apoptosis of skin cells interfering with the protective Langerhans cell–keratinocytes axis [25]. More recently, these processes have been shown to be mediated by keratinocytes and dendritic cells in non-lesional skin lesions [26].

The available clinical trial data regarding musculoskeletal manifestations demonstrate an improvement in the phase IIb trial MUSE [17]. Of those patients with ≥8 tender and swollen joints, the percentual difference at 24 weeks of ANI responders (n = 46) was 8.9% (*p* = 0.351) compared to PBO (n = 37) at a dose of 300 mg. However, at 52 weeks, 48.6% (PBO) vs. 69.6% (ANI 300 mg) of patients responded (percentual difference of 21%, *p* = 0.038) [17]. In the TULIP-1 trial, 22/68 (32%) PBO-treated patients versus 33/70 (47%) ANI-treated patients had a ≥50% reduction in active joints at 52 weeks [2]. Lastly, in the TULIP-2 trial, 42.4% of ANI-treated patients with ≥6 swollen or tender joints had a non-statistically significant response compared to PBO (37.5%, *p* = 0.55) [1].

Taken together, numerically more patients with at least six swollen or tender joints treated with ANI had a 50% or greater reduction from baseline to 52 weeks in the swollen joint count (57% vs. 46%, *p* = 0.027), and a 50% vs. 43% reduction in the tender joint count (*p* = 0.095). Thus, ANI seems to lead to an improvement of joint manifestations in a relevant proportion of patients after 52 weeks.

## 5. Conclusions

Overall, biological therapies with different mechanisms of action are sparse in SLE, and ANI is only the second approved biological therapy after BEL. There is vast experience with BEL as an add-on therapy for non-renal and, more recently, also LN [27]. Anifrolumab’s place in therapeutic algorithms has not been determined as of yet. However, our early clinical experience and review of the available clinical trial data show promising and rapid results for (refractory) cutaneous and joint manifestations in SLE, suggesting a potential domain-based approach in the near future.

## Figures and Tables

**Figure 1 jcm-11-03449-f001:**
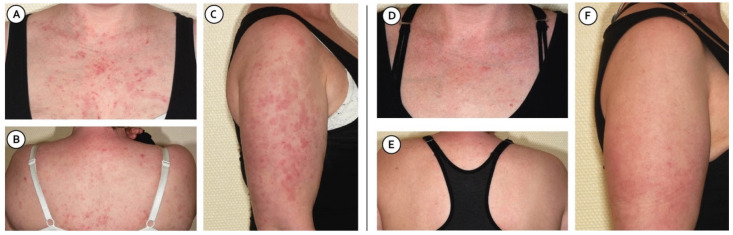
Cutaneous manifestations of the patient before (**A**–**C**) and eight weeks after the initiation of anifrolumab treatment (**D**–**F**).

**Figure 2 jcm-11-03449-f002:**
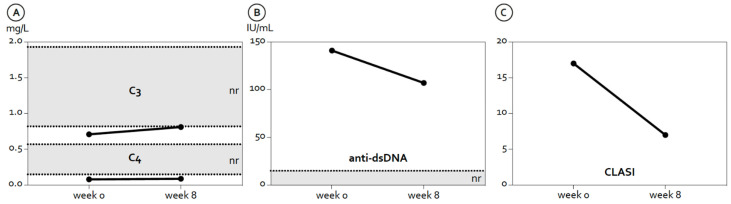
Changes in serologic parameters and the disease activity at baseline and after eight weeks. (**A**) Complement factors C3 and C4. (**B**) Anti-double stranded (ds) DNA levels. (**C**) Cutaneous lupus activity and severity index (CLASI).

**Figure 4 jcm-11-03449-f004:**
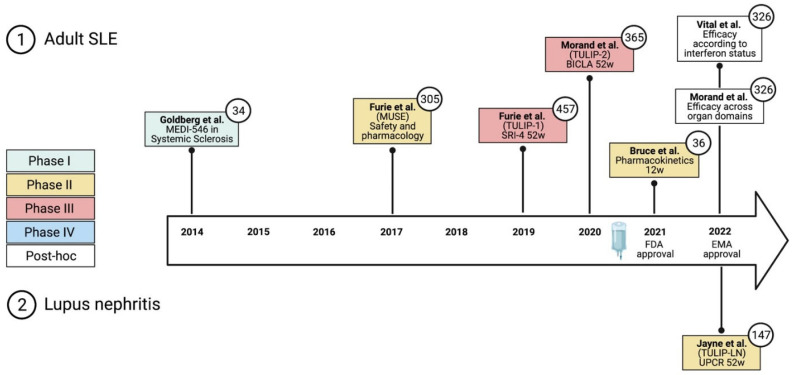
Timeline of major clinical trials, primary outcome measures, and authorization of anifrolumab [1,2,3,4,5,6,7,8]. BICLA, British Isles Lupus Assessment Group-based composite lupus assessment; EMA, European Medicines Agency; FDA, Federal Drug Agency; SC, subcutaneous; SRI-4, systemic lupus erythematosus responder index-4; UPCR, urine protein–creatinine ratio; w, weeks. Numbers in circles denote the number of participants. Created with biorender.com.

## Data Availability

Not applicable.

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
