# Peer review of "Rapid Response of Refractory Systemic Lupus Erythematosus Skin Manifestations to Anifrolumab—A Case-Based Review of Clinical Trial Data Suggesting a Domain-Based Therapeutic Approach"

_jcm, 2022, doi:10.3390/jcm11123449_

Round 1

Reviewer 1 Report

The authors refers to cutaneous and joints manifestations in SLE patients in the conclusion part in line 231. The authors discuss about skin manifestations in discussion part in detail. But, there are no description about joints manifestations in the discussion part. The authors should discuss about results of joint manifestations in the clinical trials and other studies.
To add above description, I think the authors can delete unnecessary explanation of CLASI  from line 187 to 201. 

Author Response

The authors refers to cutaneous and joints manifestations in SLE patients in the conclusion part in line 231. The authors discuss about skin manifestations in discussion part in detail. But, there are no description about joints manifestations in the discussion part. The authors should discuss about results of joint manifestations in the clinical trials and other studies.

R/ We agree with the reviewer and added a paragraph (lines 236 to 249) discussing effects on musculoskeletal manifestations to the discussion section as requested. We did not add numeric data in a table because this paper focuses on skin manifestations.

To add above description, I think the authors can delete unnecessary explanation of CLASI  from line 187 to 201. 

R/ We thank the review for his suggestion. Nevertheless, we would prefer to keep the discussion on the BICLA and SRI-4 endpoints (lines 187-196), as this has been a matter of discussion after publication of the TULIP trials and not every reader may be familiar with it. Further, not every general Rheumatologist may be aware of the domains of the CLASI, which, in our view, is crucial to understanding Anifrolumab’s effects (and the interpretation of the published and ongoing trials). If the reviewer does not entirely oppose, we would prefer to keep the CLASI description in the discussion, but we are happy to leave it to the editor handling this paper to decide whether to keep or delete.

Reviewer 2 Report

The Authors described an interesting case showing quick improvement of skin involvement after anifrolumab administration.

Although, from a clinical point of view the skin manifestations are not of the major significance in assessment of mortality risk in lupus patients (e.g. lupus nephritis or central nervous system involvement play a leading role).

However, from a patient's point of view, skin manifestations are of great importance, taking into account individual and psychosocial aspects. In this regard, the case report brings important information about clinical effectiveness of anifrolumab.

Still, further observations, outside the clinical trials, are necessary to strengthen the knowledge about anifrolumab efficacy in high activity lupus and its role in organ manifestations improvement (e.g. kidney involvement). But, as the first description of the quick response to this treatment, undoubtedly, it will be useful for further studies.

In my opinion, the description of the case and the reference to the literature are complete and correct.

Author Response

The Authors described an interesting case showing quick improvement of skin involvement after anifrolumab administration.

Although, from a clinical point of view the skin manifestations are not of the major significance in assessment of mortality risk in lupus patients (e.g. lupus nephritis or central nervous system involvement play a leading role).

However, from a patient's point of view, skin manifestations are of great importance, taking into account individual and psychosocial aspects. In this regard, the case report brings important information about clinical effectiveness of anifrolumab.

Still, further observations, outside the clinical trials, are necessary to strengthen the knowledge about anifrolumab efficacy in high activity lupus and its role in organ manifestations improvement (e.g. kidney involvement). But, as the first description of the quick response to this treatment, undoubtedly, it will be useful for further studies.

In my opinion, the description of the case and the reference to the literature are complete and correct.

R/ we thank the reviewer for her or his positive assessment of our report.

Reviewer 3 Report

This is a well-written and timely manuscript demonstrating the potential role of anifrolumab (ANI) as a target therapy for skin manifestations in refractory lupus. My only suggestion for authors to clarify that the patient was on combination therapy including belimumab for 12months prior switching to ANI in the abstract.

Author Response

This is a well-written and timely manuscript demonstrating the potential role of anifrolumab (ANI) as a target therapy for skin manifestations in refractory lupus. My only suggestion for authors to clarify that the patient was on combination therapy including belimumab for 12months prior switching to ANI in the abstract.

R/ We thank the reviewer for his or her positive assessment of our paper and the suggestion. We clarified this point in the abstract. We changed the sentence to: “Here, we report a case of SLE with a predominant and refractory cutaneous manifestation despite combination treatment with glucocorticoids, hydroxychloroquine, mycophenolate mofetil, and belimumab, which had been present for more than 12 months.“

Round 2

Reviewer 1 Report

The authors added an explanation of the efficacy for joint manifestations of anifrolumab in clinical trials. And the author believes that description of CLASI is very important. Respecting the author's ideas, I think the current form is OK.